# Integrated Planning: Towards a Mutually Inclusive Approach to Infrastructure Planning and Design

**Dario Hernan Schoulund [1,\*], Carlos Alberto Amura [2] and Karina Landman [3]**

[1] Department of Architecture, Faculty of Engineering, Built Environment and Information Technology, Hatfield Campus, University of Pretoria, Pretoria 0002, South Africa
[2] Department of Constructions and Structures, School of Civil Engineering, Engineering Faculty, Las Heras Campus, University of Buenos Aires, Buenos Aires 1126, Argentina; ingamura@fi.uba.ar
[3] Department of Town and Regional Planning, Faculty of Engineering, Built Environment and Information Technology, Hatfield Campus, University of Pretoria, Pretoria 0002, South Africa; karina.landman@up.ac.za
\* Correspondence: Dario.schoulund@up.ac.za; Tel.: +27-82299-8947

**Abstract:** Increasingly independent fields of specialization, civil engineering, and urban design find themselves practicing in isolation on the same urban issues. The result surfaces on the relative qualities of public spaces: projects that are functionally successful but spatially poor, and vice versa This is critical in the global south, where infrastructure is prioritized, and politicized, as the key driver of change but often heedless of spatial consequences. The present study explores the dynamics of integration between logics arising from technical and spatial fields, and the planning processes under which such integration is feasible. An urban design/infrastructural project in Argentina, stalled for more than two decades under regulatory policies, was selected as a case study. An overview and background of the adopted planning/design methodologies are followed by a structural/spatial analysis, focusing on type, logistics, and construction on the one hand, and on indicators of successful public spaces on the other: access, uses, comfort and image. Aspects that a priori appeared as inevitable compromises found a common, but the critically logical ground in which urban and structural thinking complemented each other. More than a functional asset, infrastructure presents an opportunity to re-think the future of the built environment as a typology that could be conceived, designed and evaluated, on the same terms as successful public spaces.

**Keywords:** integrated spatial planning; urban infrastructure; structural design; urban design

## 1. Introduction

There is a tendency to conceive of projects in terms of very distinct and often separated phases. This is especially true of urban infrastructure projects. While this may assist to demarcate the role of various professionals involved, it also tends to create a silo approach [1] and an unhealthy separation of issues related to spatial design on the one hand and technical considerations on the other. Such a separation has often adversely influenced the quality of the built environment and the potential for quality of life in cities. This article focuses on the role of integrated spatial planning and design to align the contributions from different professional fields and move towards more integrated urbanism with a stronger sense of place. This is illustrated through an urban design/infrastructural project in Argentina developed by the Municipality of Villa Carlos Paz. The objective was to bridge the development divide between the two central precincts (east and west) separated by the San Antonio River. The project presents an attempt to integrate urban and structural design as part of a simultaneous process and not distinct stages of development. This process is discussed through an introduction of the project context and planning/design process, followed by a detailed discussion of the choice and nature of the structure and the spatial solutions. The paper argues for a trans-disciplinary perspective towards infrastructure and advocates that structural and spatial considerations should not be mutually

exclusive but should be aligned to enhance both the efficiency and the experience of urban infrastructure projects.

### 1.1. A Spatial or Functional Approach: Mutually Exclusive?

The current 'infrastructure turn', where emerging global infrastructure practices are supported by a new set of discursive, political, and technical arrangements, tends to displace and override spatial planning practice [2]. In addition, fragmented knowledge of infrastructure across different disciplines compromises the development of robust planning strategies [3]. Even when the bond between infrastructures and cities is tight as ever, the relationship between planning/urban design and infrastructure remains noncomprehensive and nonstrategic [4]. Part of the problem lies in that infrastructure is often framed as an abstraction [5], prioritized—and politicized—as an essential driver of change, but heedless of spatial implications.

The institutional and epistemological gaps between spatial and technical disciplines often reflect on regulatory planning approaches, and consequently in the quality of the urban experience itself. On the one hand, the planning and urban design professions' interest in infrastructure has declined in deference to specialists [4], leaving key roles to professions mostly concerned with efficiencies, such as engineers, financers, managers, and public work officials. On the other hand, infrastructures' technical and organizational aspects are unappealing to the architect, as they require a "different habit of mind about design" [6] (p. 264). Thus, architects and urban designers have focused overwhelmingly on the design of spaces within envelopes rather than the networked infrastructures that bind and configure them [7] (p. 18), giving rise to infrastructural urbanism [8], where reductionist spatial engineering asserts itself.

While planning tends to deal more with the two-dimensional spatial organization and regulation of the city, urban design focuses on the three-dimensional quality and organization of the built environment. Integrated planning, or what we refer to as inclusive planning, presents an alternative defined by horizontal and vertical integration of governance levels. As opposed to regulatory planning, integrated planning relies on appropriate methodologies, management, public involvement, and coordination that respond and adapt to local conditions [9]. This necessitates planners to move beyond their traditional functions to a more communicating and mediating role [10]; or to what has been called participatory design [11] or co-design [12]. These approaches emphasize the role of the planner/urban designer in working with relevant stakeholders to encourage a more positive and sustainable outcome. Although participatory planning has shown to be beneficial, developers and planners often neglect this and fail to involve all the stakeholders [13]. At the same time, it reflects the increasing focus in urban design theory and practice to acknowledge urban design as both product and process [14–16], where it is recognized that the process plays a very important part to create quality places. Four process dimensions, namely design, development, space in use, and management play an important role to shape place over time [15].

Due to logistical challenges, however, these processes are not exempt from risks: Holden [17] warns that preconditions for integration must be present across normative dimensions. Similarly, the 'tree planning approach' [18] proves to be a barrier to institutional integration. In addition, there is often not enough time spent in the planning or design phase, as projects need to be delivered very rapidly [19]. Another concern relates to the measurement of design outcomes or performance measurement [19,20]. Some of these challenges are evident in developing countries such as South Africa, where there is a lack of integration between various departments and levels of governance [21].

South America also struggles with these challenges, resulting primarily from disjointed management and less so from professional capacity [22]. Urban governance is a major challenge in itself [23], characterized by an abundance of overlapping authorities, competing departments, and uncoordinated efforts and policies. The absence of proper reviewing mechanisms, like design review in the UK [24], further contributes to

the 'end product oriented' approach. Additionally, urban design, and its potential coordinating role [25], remain a weak discipline in Argentina, often associated with exclusive developments [26], or in cases seen as a luxury [27].

In the selected case, an approach that focused on participation among departments and stakeholders was used as an alternative to hitherto unfruitful proposals under regulatory approaches. Admittedly, "negotiation, as much as collaboration, was expected" [28]. This analysis illustrates the dynamics of a process where urban design and structural engineering found a common ground, and the benefits of a broad base professional/public forum that ultimately defined a feasible proposal.

### 1.2. Project Background

The project is located in the town of Villa Carlos Paz (population 90,000), Córdoba province, Argentina (Figure 1). This town is the second most touristic place in the country, with occupancies often reaching as many as 1 million visitors during the high season (January and February). The layout of the town can be described as two similar-sized urban areas divided by the San Roque Dam and the San Antonio River. The river delineates the two halves of the downtown area, connected by the 1889 bridge ('old bride') (Figures 2 and 3). The downtown area was originally established on the west bank, but over the years expanded to the east bank, where it thrived, overshadowing the former west-central precinct.

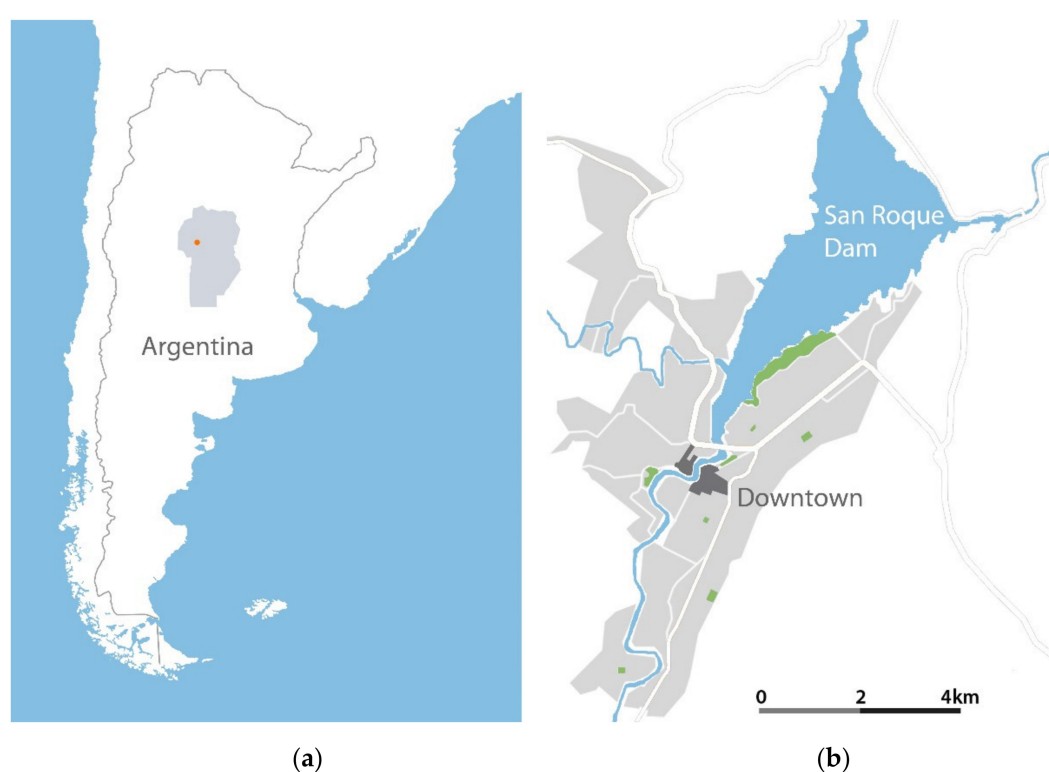

(**a**)  (**b**)

**Figure 1.** Location maps. (**a**) Argentina and Cordoba Province. (**b**) Villa Carlos Paz, the San Roque Dam, and the Downtown area on both sides of the San Antonio River.

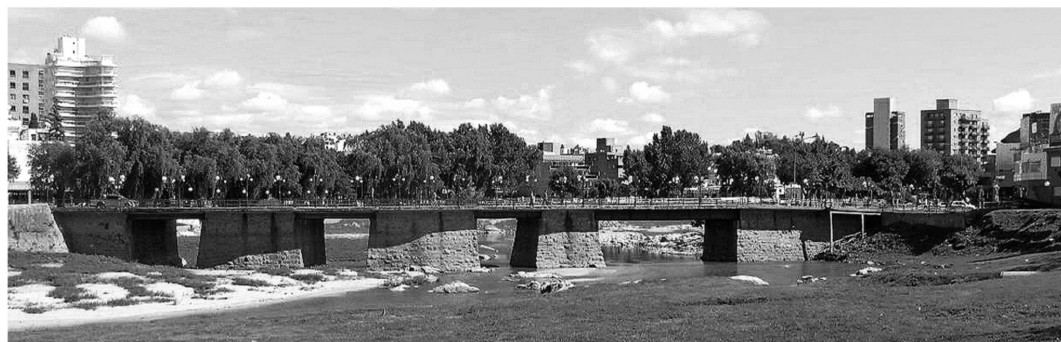

**Figure 2.** The existing 1889 bridge, its supporting piles account for 45% of the drainage section. (Credit: Reproduced with permission from the author, Eng. Gerónimo Cáffaro).

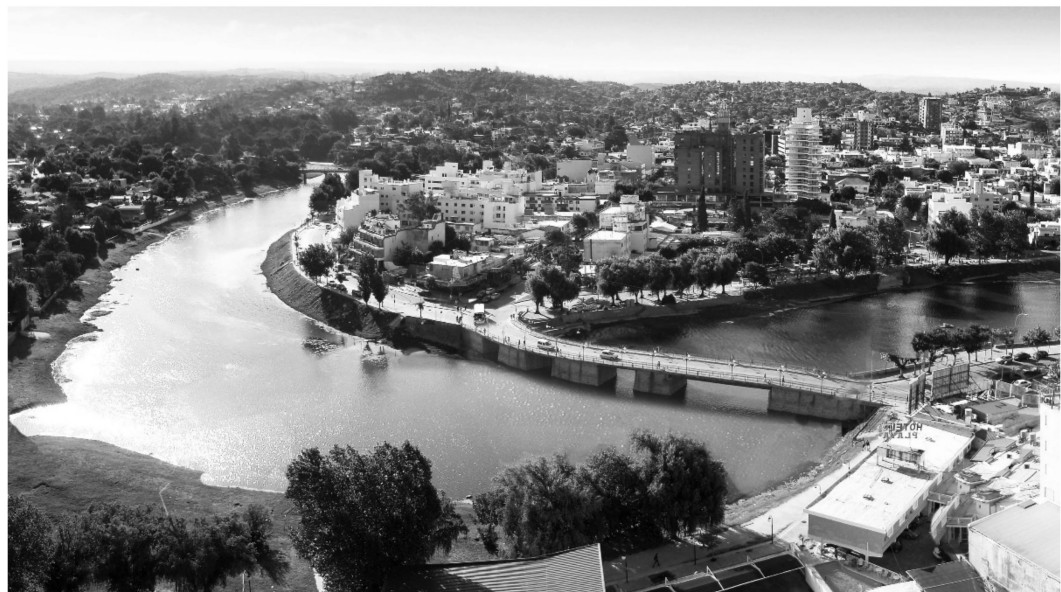

**Figure 3.** Aerial view taken from the east precinct, showing the San Antonio River and the 1889 bridge. Mixed-use, mostly retail and residential, occurs in low to medium densities.

The general objective of this project was to bridge the increasingly marked development divide between the two central precincts. The need was identified during the 90s when the town grew rapidly in popularity. Although several proposals were developed in the subsequent years, they did not meet enough political/financial traction, or public acceptance [28]. Additionally, these proposals focused on replacing the existing old bridge with wider and modern structures. Even when the old bridge has proven structurally unsound and hydraulically problematic, its replacement remains a sensitive heritage issue. These aspects compounded, resulting in the project stalling for more than two decades.

### 1.3. Planning Methodology

From 2014, the municipal planning department managed an integrated process. The methodology aligns to some degree with that proposed by Yigitcanlar and Teriman [10], consisting of a series of re-evaluative steps that belong to two distinct phases: definition and confirmation (Figure 4). In the definition phase, the integration places particular emphasis on 'participatory design' as an extension of participatory mapping techniques. A neutral-role format was employed for a series of design workshops that culminated in the urban vision and later on the design parameters of the two proposed bridges, promenades, and public spaces. The rationale behind focused on removing the boundaries among the three main stages of regulatory planning (Figure 5), and the boundaries of professional

fields. In this manner, the identification of the issue, the 'what-to-do part' of the process, benefited from a broader perspective in which the wealth of local knowledge formed a key contribution.

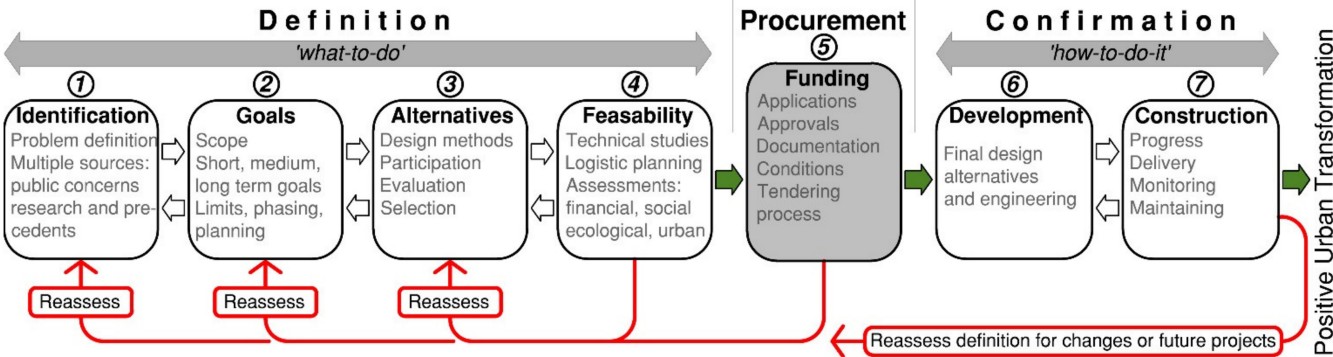

**Figure 4.** The integrated model was employed, placing particular emphasis on the definition.

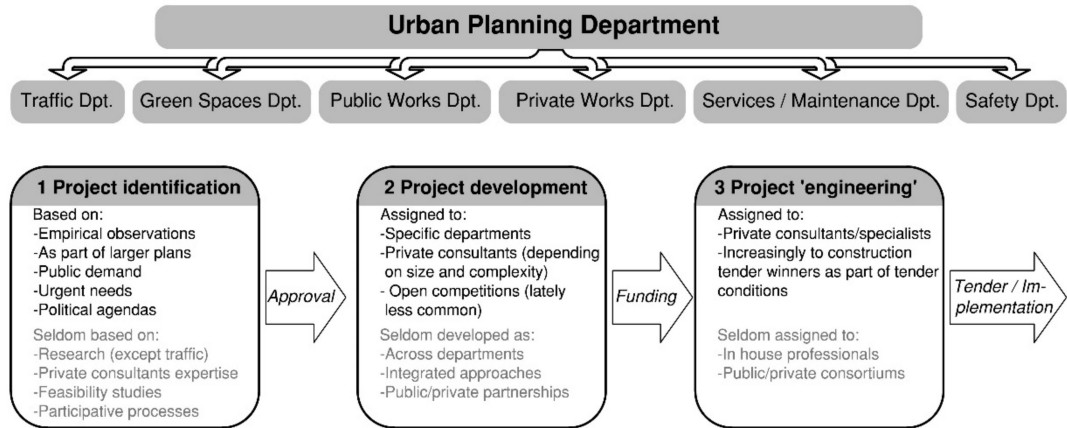

**Figure 5.** Three main stages of regulatory planning are typically assigned to specific departments. The many variables make it difficult, if not impossible, to expect a resulting integrated proposal.

The first three planning steps of definition, comprised of a series of information and design workshops, where public officials, private consultants, and the public, in general, took part. The focus was on reframing the project from an engineering issue (conceived and developed by specialists) to an urban design issue (conceived by many). Stakeholders also included urbanists, project managers, architects, landscape architects, engineers (traffic, civil, and hydraulic), politicians, and historians. Subsequently, during the confirmation phase, the focus was specifically on urban and structural design, the 'how-to-do-it' part of the process. Facilitators encouraged participation in all aspects of design solutions: engineers were encouraged to design, and share ideas with the 'designers' (urban designers/architects), in turn, designers were prompted to conceive and participate in technical solutions. The exchange of ideas was at the core of the process, and an experimental element, which in regulatory terms is considered a risk, was embraced with motivation.

## 2. Methods

A case study research approach was employed, together with a documentary data collection method. An overview of the project and a background of the adopted planning/design methodologies places the process within integrated planning boundaries. To establish the relationship between technical efficiency and spatial considerations, the

main structural features (type, geometry, and implemented solutions) were analyzed with criteria of successful public spaces and a range of spatial methodologies for understanding public space and the performance of the built environment. There has been extensive research on what constitutes successful public spaces or precincts. For example, Montgomery notes that the components contributing to a sense of place relate to activity (including access, diversity, vitality, use, etc.), form (including permeability, landscaping, and scale), and image (including symbolism and memory) [29]. The Project for Public Spaces' Place Diagram highlights five key qualities, namely sociability, uses and activities, access and linkages, comfort, and image [30]. Carmona identified 10 aspects of successful spaces: evolving, diverse, free, delineated, engaging, meaningful, social, balanced, comfortable, and robust [15]. These were reduced to four main indicators for the consideration of spatial quality, namely accessibility, uses, comfort, and image.

The evaluation of proposals/alternatives centered on structural and spatial qualities, followed in order of importance by management/logistical aspects, and lastly by financial considerations. Comparative charts were used as an indicative tool to evaluate and compare these aspects (Figure 6). The intention was to identify and combine the benefits of alternatives. In this simple tool, the criteria for structural design and the desired spatial qualities were outlined. Budget played no significant role, which resulted in design flexibility, variety, and choice. This approach was possible since budget allocations to (small) municipalities are irrespective of specific cost estimates but dependent on the project's relevance.

| Alternative | Structural | | | Spatial | | | | Management Logistics | Financials | Score |
|---|---|---|---|---|---|---|---|---|---|---|
| | Logistics | Construction | Type | Access | Uses | Comfort | Impact | | | |
| | local sourcing import items timeframes flood seasons | hidraulic feasability geological feasability profesional and work-force capacity | clarity definition image | connected welcoming safe convenient | diversity 24 hours democratic celebratory | walkable clean universal access inviting | real state commerce historic natural touristic | Implementation timeframes / political cycles funding opportunities Affected areas and upgrades | % ratio import/local actual price ancillary works downtime impact costs | |
| # | | | | | | | | | | |
| # | | | | | | | | | | |

**Figure 6.** Comparative charts were used to evaluate alternatives, as an indicative tool only.

Accessibility was measured through pedestrian counting recorded on January 2018, 2019, and 2020, during the most popular weekend of the high season. Comfort and image were recorded through a documentary data collection method in which publications concerning the project (broadcast, online publications, paper publications) were analyzed for the same period. Finally, direct observations attempted to connect the various uses to key structural/spatial features. This reflects the importance of trying to measure the performance of the built environment in various ways [20,31], despite the difficulties associated with such performance measurements [32].

### 3. Results

#### 3.1. Urban Vision

The first collective outcome defined the urban vision: a staged construction plan in which the old bridge was not affected in principle. The sequence entitled: (1) building of a pedestrian bridge (completed), (2) building of a mixed bridge of four traffic lanes, and (3) removing of the old bridge. The advantages of this plan are numerous: first, the three phases are independent, secondly, the decision to remove the old bridge—a sensitive point—could be delayed until consensus is reached or be left untouched, and lastly, the funding procurement model resulted in better chances of adjudication. The vision for the area involves significant public space reconfiguration (roads, green spaces, promenades, and sidewalks), leading to a more pedestrian-friendly environment (Figure 7).

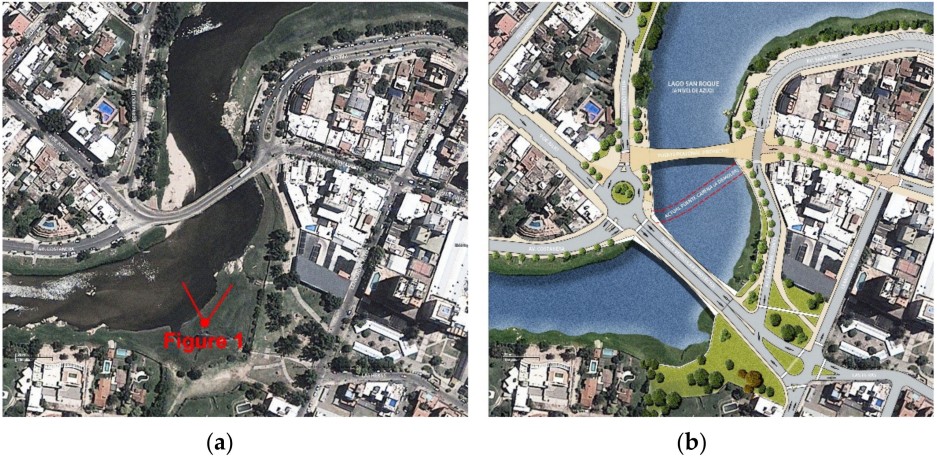

|  (a)  |  (b)  |

**Figure 7.** (**a**) The downtown area connected by the existing bridge of 1889. (**b**) The urban vision shows substantial public space reconfiguration, a proposed mixed bridge (bottom), and the implemented pedestrian bridge (top). (Credit: Provided by the planning department of Villa Carlos Paz (public domain)).

### 3.2. Pedestrian Bridge

The first project set for implementation, the pedestrian bridge, was envisioned as a public space across the San Antonio River. Yet the site constraints, seismic and flooding conditions, and the restrictions imposed on the geometric envelope presented a considerable engineering challenge that risked overriding key spatial considerations. The design premises were defined as a set of conditions:

- A comfortable pedestrian deck of 6 m of minimum width;
- A maximum height of 12.5 m above the 100-year flood line;
- A maximum of two intermediate supports (3 spans);
- An attractive, contextual, and memorable architectural language;
- A lookout point to the surroundings.

### 3.2.1. Bridge Typology

The San Antonio River is prone to flooding for 6 months of the year. Its drainage section on the location of the bridge, measured at 100-year flood recurrence (2100 m$^3$/s), is 72 m. The metamorphic bedrock sits at $\pm6$ m below shifting layers of alluvial boulders, gravels, and sands. These two aspects challenged the practicalities of building intermediate supports. Thus, two options presented themselves: to build two intermediate supports of efficient hydrodynamic profiles (for a 3-span deck), or a single 72 m span. The construction of intermediate supports offset any apparent savings derived from the simplicity of the superstructure while adding a series of constructive and hydraulic constraints. A single 72 m span avoids interference with water flow while establishing the lowest possible level for the deck above the 100-year flood line (Figure 8). In terms of accessibility, it provides a smoother transition, allowing the use of frontal ramps instead of stairs due to the reduced depth of the deck.

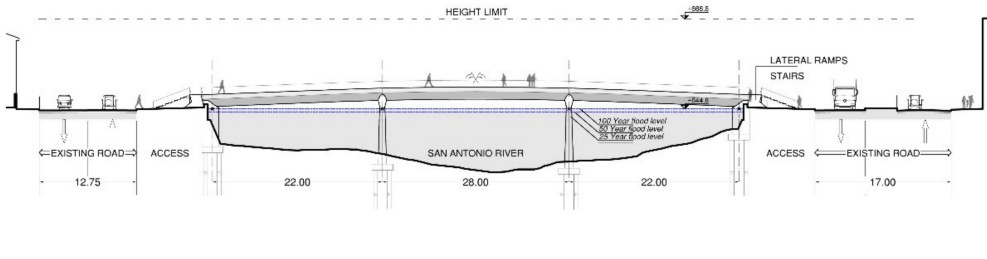

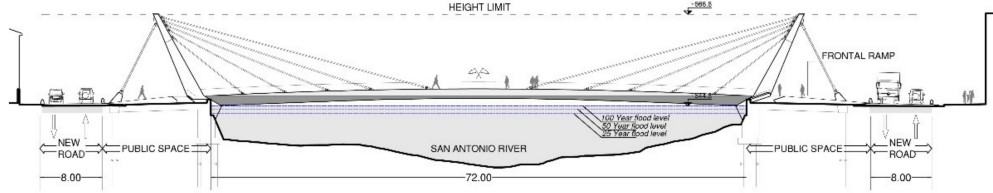

**Figure 8. Top**: early 3-span proposal: accesses were uninviting as ramps could only occur laterally. **Bottom**: selected single-span cable-stayed typology, together with new public spaces and road reconfigurations for friendlier accesses.

### 3.2.2. Structure

The bridge's substructure consists of concrete pile foundations and abutment walls, integral concrete masts, and deep active rock anchorages for backstays. The superstructure consists of a metallic web of fixed longitudinal and transversal beams coinciding with the stays' attachments (Figure 9). All stay cables are composed of parallel tendons of 7, 12 (deck), and 19 strands (backstays). Due to the very low angle of the central cables (15°) and to seismic deformations, fork anchorages at both ends were preferred to minimize fatigue at the terminal. It was therefore decided, for consistency, to utilize these attachments in all cables in what was dubbed 'honest engineering'. The bridge was modeled using the FEA system SAP2000 (V.16), considering the AASHTO LRFD Bridge Design Specification norm [33].

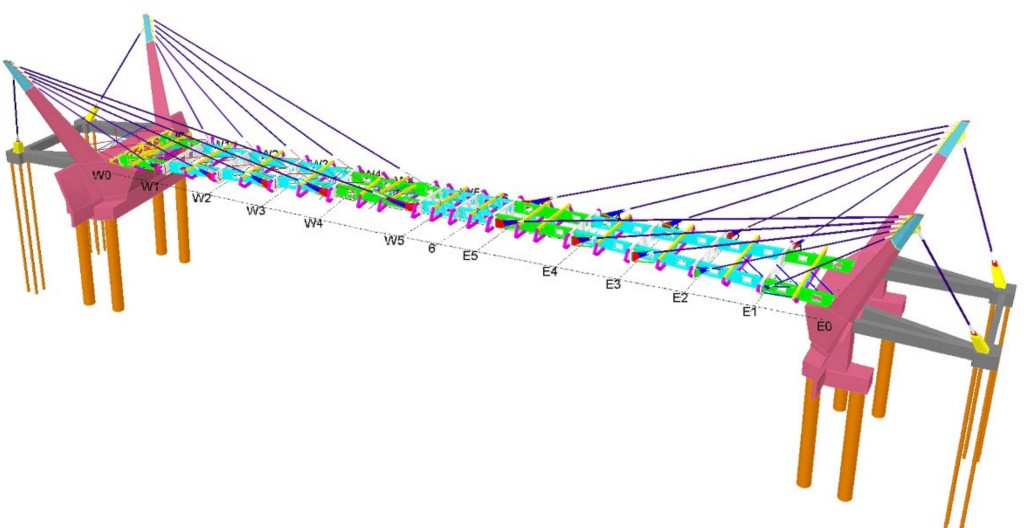

**Figure 9.** 3D BIM model of the structure.

### 3.3. Spatial Experience

The deck was to create a lookout point free of structural interference. To accentuate this premise, the central structural modules stretched from 6 m to 8 m. This small change enlarges the lookout zone from 18 m to 23 m (measured at eye level), which on a 72 m crossing represents just under a third of the total (Figure 10). Structurally, the larger

modules coincide with the lighter zone of the deck; the modules are longer but narrower. The area is further celebrated by the 1m longitudinal bulge that confers a sense of mastery over the environment while preventing debris impacts during floods.

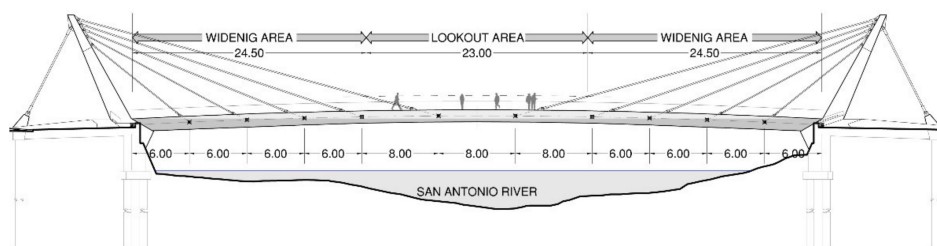

**Figure 10.** The central lookout area, higher and clear of visual interference for almost a third of the crossing.

The deck's width, first defined at 6mts, offered a generous crossing but created a blunt transition to the access public spaces. To address this point, the entry areas widened to 11.4 m, in this way, the deck 'opens' to the city together with the outward leaning masts. This geometry creates zones within the deck in which users can linger without interfering with the central flow. A variety of activities is then encouraged in line with diverse public spaces. Structurally, the extra mass of the enlarged deck coincides with the more efficient stays of greater vertical components; the logic of the load diagram matches the geometry of the deck (Figures 11 and 12).

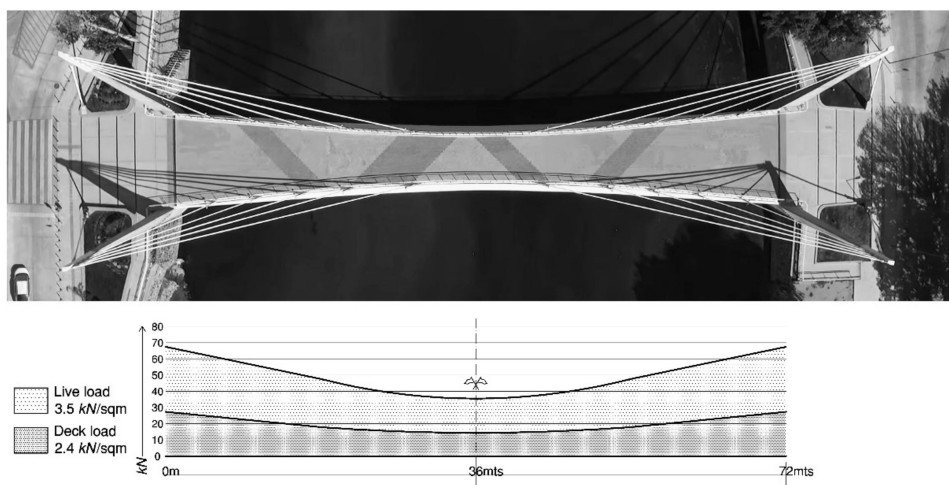

**Figure 11. Top**: the deck geometry in the plan widens from 6mts at the center to 11.4 m at the entry points. **Bottom**: the load diagram places most of the mass closer to the abutments and onto the more efficient stays.

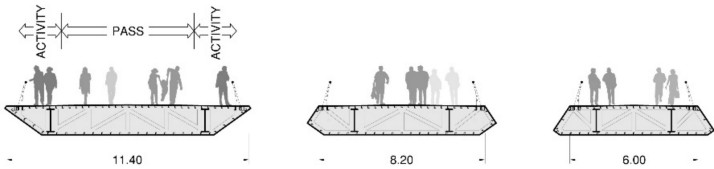

**Figure 12.** Deck's cross-sections at different instances: from left to right, entry points, quarter of span, and at the center.

The concrete masts lean backward (29°) and outwards (18°) in permanent flexion. They are integral to the abutments and blend in geometrically with the adjacent stone

retaining walls. The failure of one backstay/anchorage was modelled as a worst-case scenario; in such an event, the leaning concrete mass of each mast (42.5 metric tons) plays an important balancing role. These aspects align to open the entry areas: the masts, with their upward and outward gesture resemble 'open arms' (Figures 13 and 14). This also highlights the importance of spatial experience and symbolic interpretation of the structure.

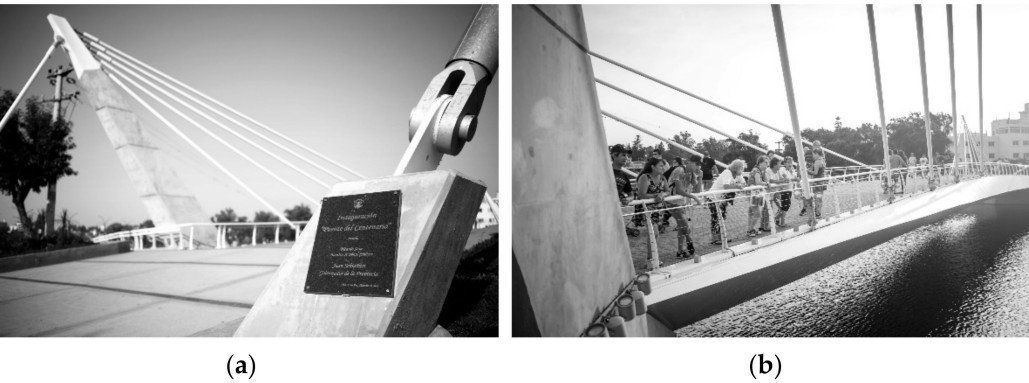

(**a**)                    (**b**)

**Figure 13.** (**a**) Exposed stay attachments as 'honest' engineering themes. (**b**) Various activities were observed in the wider areas of the deck, without interfering with the central pedestrian flow.

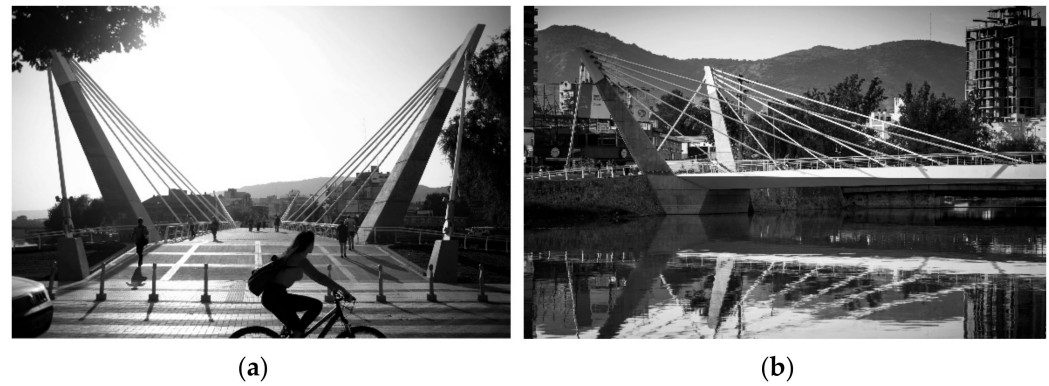

(**a**)                    (**b**)

**Figure 14.** (**a**) Masts raising upwards and outwards resemble welcoming 'open arms'. (**b**) Masts, abutment walls, and foundations form an integral element.

### 3.3.1. Usage, Access, Comfort, and Image

Usage was registered through a comparative pedestrian counting (full crossings) during the most popular weekend of the high season; first in January 2018, then in January 2019, and finally in January 2020 (January 2021 was omitted due to lockdown). The high season was selected since a requirement of the project was to become an 'attraction'. We employed two assistants per abutment: 4 in 2018 and 2019 (old bridge), and 8 in 2020 (both bridges), discreetly stationed on each sidewalk/side, counting only exiting pedestrians. The counting took place from 7:00 a.m. until 12:00 a.m. for three consecutive days using a mobile counting application. All nine days' results are shown in Figure 15.

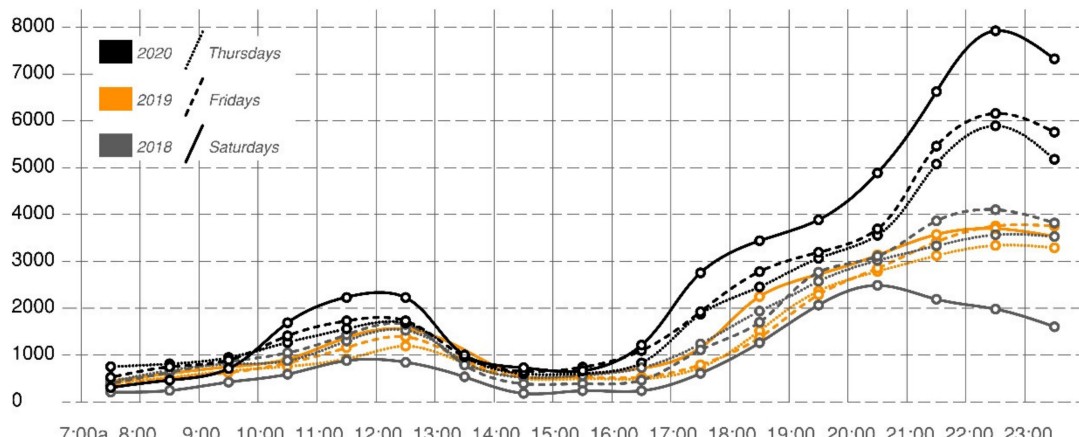

| Year | Total crossings (3 days) | ⬆ To old centre | ⬇ To new centre | Using old bridge | Using new bridge | % + -<br>(from previous year) |
|---|---|---|---|---|---|---|
| 2018 | 72,510 | 34,665 (48%) | 37,845 (52%) | | | |
| 2019 | 78,122 | 38,068 (49%) | 40,054 (51%) | | | + 7.7% |
| 2020 | 124,619 | 55,630 (45%) | 68,989 (55%) | 44,356 (36%) | 80,263 (64%) | + 59.5 % |

**Figure 15.** Periods of activity relate to summer temperatures, decreasing during the hot afternoons and increasing during the nights, and well into the next day.

Pedestrian activity increased for 2020, peaking consistently on Fridays and Saturdays, except for the year 2018 (Saturday) due to unstable weather (drizzling but pleasant temperature). Interestingly, not all users crossed the bridge; many would enter, take photographs, observe, linger and return. We then employed one assistant per abutment to discount "visitors" from "crossers"; therefore, more people than counted visited the immediacy of the project.

### 3.3.2. Documentary Data

Reporting of the project in public media was classified as negative, neutral, and positive. On the negative aspect, the project was criticized for not being an essential priority in a town that lacks basic services in some areas [18], and for traffic and services disruptions spanning two years [34,35] (Figure 16). The negative phase moved into the expectation phase before completion, where reporting shifted to the nature of the project, its scale, and its meaning. Imminent inauguration, coinciding with the high season, created some positive coverage of the political dimension [36,37] (Figure 17). Finally, once inaugurated, an element of novelty took place. Neutral reports that indirectly included the project as a referential place started to appear, and several events and small rites were observed: engagements, wedding photography, lovelocks, T.V. interviews, promotions, and a public protest (Figure 18).

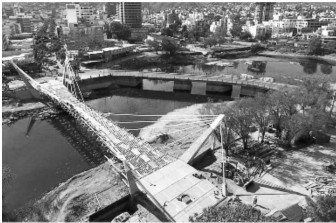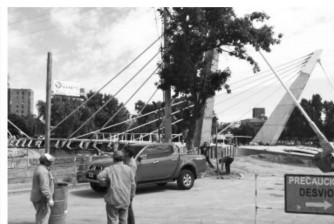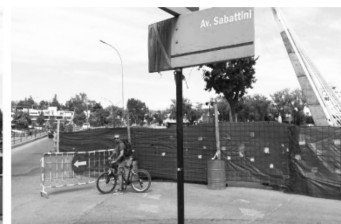

**Figure 16.** (left to right) Road disruption during construction: the main source of negative reporting, affecting pedestrian and bicycle traffic as well as key trunk services for almost two years. (Credit: Left: From public television broadcast (Cordoba Governance); Centre and Right: reproduced with permission from the author Mr. Mario Rojas (Centedario digital publication)).

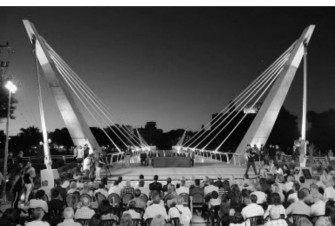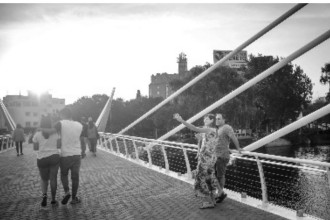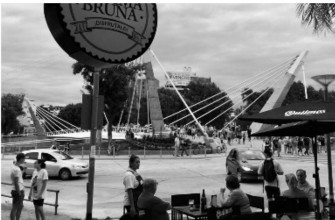

**Figure 17.** (**left**) Political promotion during the inauguration ceremony. (**center** and **right**) The bridge is used more as a public space. (Credit: Left: From public television broadcast (Cordoba Governance)).

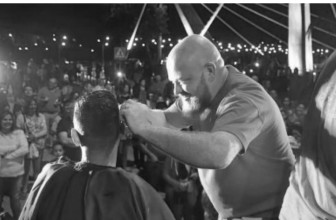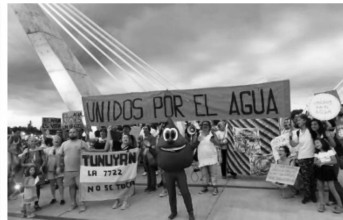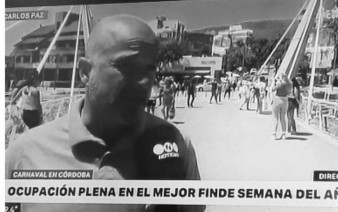

**Figure 18.** The project as a democratic venue: (**left**) Hairdressers' marathon (charity event). (**center**) Public protest on water management. (**right**) T.V. interviews using the bridge as a referential place. (Credit: Left and Center: reproduced with permission from the author, Mr. Mario Rojas; Right: Still from public television broadcast (Cordoba Governance)).

*3.4. Fusing Structural and Spatial Considerations*

Almost in all cases, the preferred structural solutions were also the higher scoring ones in terms of spatial considerations. Good structural solutions tended to be financially reasonable: a minor extra cost represented a significant increase in spatial value. This point is particularly clear on the new public spaces: they play key roles in calming traffic, in defining a safe pedestrian area, and in protecting the backstays against potential impacts. These spaces cannot be separated from the bridge: they are part of the structure (Figure 19). The same logic extends to material selection: the green islands' edges are designed as continuous benches, but they are constructed as reinforced concrete barriers.

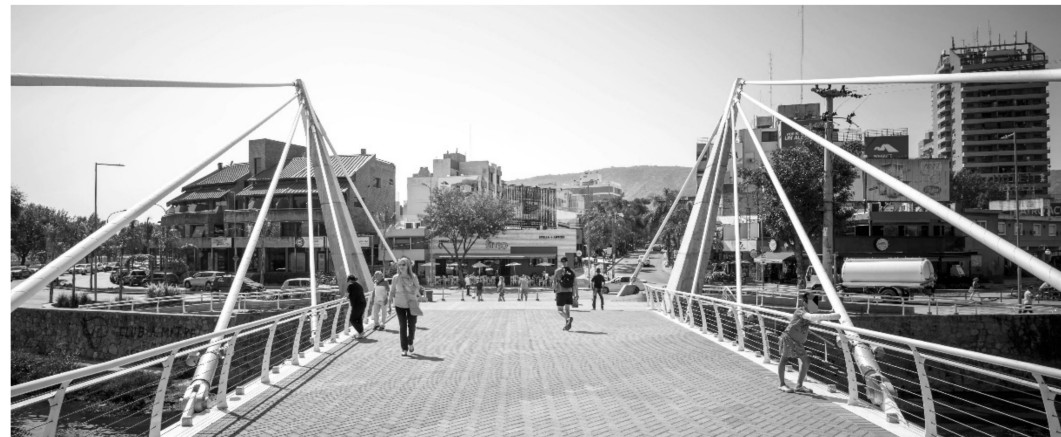

**Figure 19.** Pedestrian view looking towards the east precinct from $\frac{1}{4}$ of the span.

This common ground between urban and structural design presented itself on each composing aspect, from major features to the detail of smaller parts. Similarly, the larger area of the project, which involved the design of promenades, parking bays, a bus stop, road/traffic reconfigurations, light fittings, and green spaces, responded—and was guided—by the same spatial/functional premise (Figures 20 and 21). It, therefore, represents a fusion of spatial and structural considerations to become more than just a bridge but a place of encounter, crossing, and reflection.

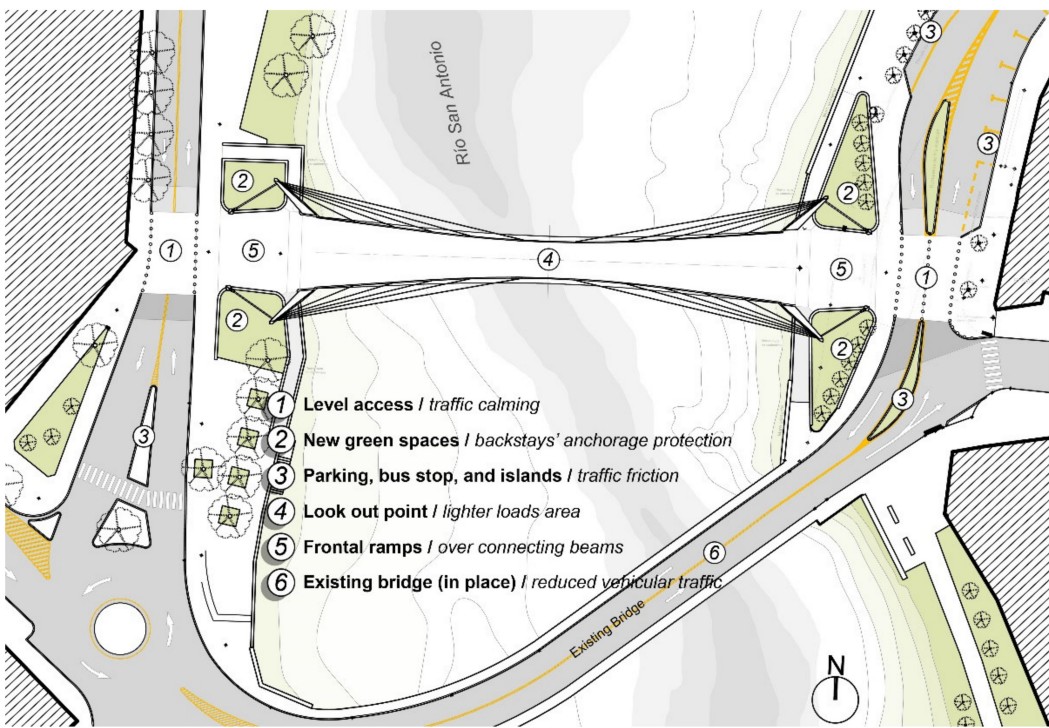

**Figure 20.** Ancillary works included traffic reconfiguration, parking, pedestrian crossings, public spaces, and promenades.

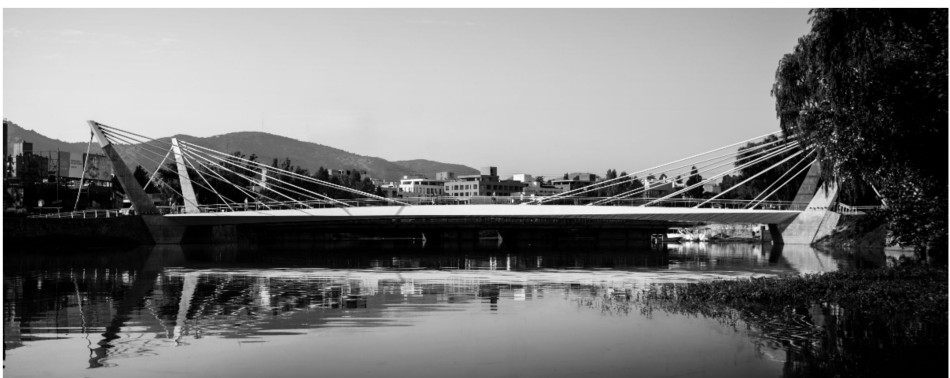

**Figure 21.** The completed project was inaugurated in 2020, with the 'old bridge' partially visible behind (upstream).

## 4. Discussion

### 4.1. Towards a Mutually Inclusive Approach to Infrastructure Planning and Design

Despite its short normal operational lifespan, essentially less than 3 months before the COVID-19 lockdown, several interesting events occurred in and around the project. At the same time, increased pedestrian traffic aligns with the main intention of equalizing both halves of the central precinct: to bring life to the west bank. Two logics needed to be fulfilled: the logic of urban design as manifested in successful public spaces and the logic of efficient structural engineering. In this project, these arguments have proven to be the same, suggesting that at no point does any design aspect reflect a trade-off or a compromise. Sound structural solutions tended to be superior spatial solutions. This approach in which infrastructure is public space requires a shift in perceptions: it implies that public space, instead of infrastructure [38], is the medium that binds the city together, defined by the constant combination of multiple components, including infrastructure. Therefore, the importance of aligning and fusing structural efficiency and spatial considerations should benefit under the more detailed understanding of urban design.

In the present case, the importance of coordination at the Municipal level, where professional, scientific, and educational institutions participate, was central [9]. Secondly, participatory design [10] became the vehicle through which design professionals sourced contextual creative material. This also highlights the value of participatory design; both as a means to give voice to the users of space but also as a reflective practice [11]. It also echoes the importance of working with the place to find the value-adding role of each element and member of the system towards evolutionary sustainability [39]. Lastly, due to the self-evaluating possibilities embedded within the process, changes and reconsiderations were possible, even during construction, which emphasizes the important role of adaptive planning and design to enhance the quality of the built environment.

### 4.2. Aligning Technical and Spatial Thinking

For decades, authors have noted that creativity in the instruction of engineering is not encouraged [40–43]. Leonhardt [44] suggested that the profession only attracted "persons who have a talent for rationalism and logical thinking, but no sensitive feeling". However, the experience of this process tells otherwise: engineers are highly creative in technical terms and resourceful when discussing various alternatives, which is often during the early stages of project development and precisely when their participation is typically absent. Creativity is thus not only limited during training as part of a critical foundation but also systemically avoided during regulatory practice. As Christian Menn [45] points out, "The most challenging aspect of a bridge engineer's work is conceptual design". If engineers are not exposed to an ambit conducive to creative challenges, their work is relegated to the 'engineering' of a given project.

On the other hand, urban designers considered themselves designers, mostly due to their architectural backgrounds, but admittedly not trained—or interested—n technical aspects of functional typologies. As opposed to engineers, their intuition for form and balance is only superficially related to technical considerations, as it is not conceptualized from such sources. Instead, designers are particularly good at identifying avenues of creative thinking within the multiple issues touching urban projects, even political or financial for instance. In this case, designers asked the 'right' questions, which triggered engineers' creativity with multiple answers/solutions. Thus, the combination of designer and engineer is not merely a partnership set up to build; it is a method for interpreting public and political interests to their ultimate physical solution. Arguably, this project moved from being an engineering issue to an urban design one. It found no clear direction as long as it was framed within technical boundaries. Only when considered from a wider urban design perspective the appropriate answers start to present themselves; an approach to design that is "not always about meeting the exact standards and having the right answer" [46] (p. 265), but working with the story of the place and conscious interventions in the right place to create system-wide effects [39].

This reiterates the importance of not only focussing on the product of urban design but also the process. Focussing on the process has the potential to add significant value in the Global South to acknowledge the specific context through the various dimensions of placemaking, namely the spatial, procedural, and psychological dimensions [47]. The spatial dimension was reiterated through the strong visual qualities of the structure and the places created around the bridge. The procedural dimension was addressed through the participatory design process. Finally, the psychological dimension has been acknowledged through the positive responses by the users of the space. This allowed for the nuances of language to allow meaning to be relayed in particular ways unique to the context, a way to address the specific context of the Global South [48].

### 4.3. Key Considerations for Infrastructure

Whether the infrastructure is different from other types of urbanization has been questioned [2]. The relation between its engineering nature and the spatial implications on urban form, regions, or even countries, defines the concept of 'infrastructure space' [46]. This connection cannot be isolated because infrastructure is not an end in itself, nor is a spatial vision possible without its supporting networks. Yet in South America, infrastructure represents primarily provision, prioritized, and politicized at all costs above other components of the urban and natural environment. The implications, when understood, are regrettably accepted as a fact-of-life, in which "one person's infrastructure is another's person difficulty" [49]. Public space becomes an increasingly contested arena, where components deriving from different disciplines find little coordination and coherence, or no place at all.

It is, therefore, necessary to review infrastructural projects under the urban design concern. This premise underlies the present case: the structural type promotes the diversity of spatial features, but the type responds to a greater urban strategy. Such alignment between structure and space suggests that professional specialization and regulatory planning retain the underlying logic between these fields. After all, a bridge is a particular case whose landmark significances, both figurative and literal, have not changed over the years. Other typologies that had a structuring presence in the past, like water provision and distribution, are now largely backgrounded. Conversely, underground stormwater networks are surfacing with a strong image and structuring capacity. These types of interventions highlight the potential of using infrastructure to increase a sense of place and sustainability. A recent report by Cambridge University [50] (p. 13) emphasizes the role that infrastructure can play towards place-making and greater sustainability: an example from Cape Town is presented in which hard infrastructure (Bus Rapid Transit) is combined with the creation of public spaces and facilities. At another level, a greater focus on green infrastructure in cities can also support both the goals of sustainability and a sense of

place [51]. If a balanced approach to public space design prevails, one in which all its intrinsic components are encouraged to be in dialogue, we can expect to be on the right track to a more meaningful urban environment.

## 5. Conclusions

Infrastructure planning tends to focus too much on function at the expense of spatial quality. On the other hand, the planning professions tend to focus too much on regulation, while urban designers are often only fixated on the product and its aesthetic qualities. This article examined a specific urban design/infrastructure project in Argentina developed by the Municipality of Villa Carlos Paz. The project involved the building of a pedestrian bridge to address the divide between two central precincts in the city. The discussion indicated that the project did not only connect the two parts of the city but also bridged the gap between urban and structural design through the planning process, design, and implementation of a project that was, under regulatory policies, unprovable. It also illustrated the possibilities of achieving good results when the barriers of communication and coordination—a key challenge of the Global South-are displaced to create space for dialogue and collaboration: through a process-oriented approach based on inclusive planning. This involves inclusivity at three levels: (1) including a focus on both the product and process of urban design, (2) involving all the relevant professionals of the built environment in all phases of the project to allow co-design, and (3) working with municipalities, communities and the relevant stakeholders to enable a participatory design that would address the context-specific issues towards appropriate placemaking in the Global South. Restricted funding and fewer projects make it even more important than the money spent on placemaking projects in the Global South should be able to create accessible, useful, comfortable, and meaningful places.

The main role of inclusive planning centers on facilitating the alignment of design logic arising from different professional fields and on guaranteeing a positive degree of contextualization. At the same time, this approach to planning promoted the articulation of the many voices' concerns and ideas into genuine sources of creative thinking. The fusion of urban and structural design shows the potential to increase the sense of place and contribute to greater urban sustainability. What is promising, therefore, is the possibility of integrated typologies that challenge the idea of infrastructure and public space as separate components of the urban experience. Critical for the future of the built environment, such an approach could also revitalize the declining relevance of the traditional public space by attaching its design principles to the infrastructures of daily life. Given the increased importance and multiple demands placed upon urban infrastructure, integrated spatial planning represents an alternative for conceiving, developing, and enhancing projects both functionally and spatially.

**Author Contributions:** All three authors participated actively in this study. D.H.S. provided the urban design and architectural perspective from which the project emerged. C.A.A. provided the engineering counterpart, and K.L. a combination of planning framework, methodology design, and research structure. Conceptualization, D.H.S. and C.A.A.; methodology, D.H.S., C.A.A., and K.L.; software, C.A.A.; validation, D.H.S., C.A.A., and K.L.; formal analysis, D.H.S., C.A.A., and K.L.; investigation, D.H.S.; resources, D.H.S. and C.A.A.; data curation, D.H.S., C.A.A., and K.L.; writing—original draft preparation, D.H.S.; writing—review and editing, D.H.S. and K.L.; visualization, D.H.S. and C.A.A.; supervision, K.L.; project administration, D.H.S.; funding acquisition, n/a. All authors have read and agreed to the published version of the manuscript.

**Funding:** This research received no external funding.

**Institutional Review Board Statement:** Not applicable.

**Informed Consent Statement:** Not applicable.

**Data Availability Statement:** Not applicable.

**Acknowledgments:** We would like to thank the assistance of the Director of the Urban Planning Department of Villa Carlos Paz, Liliana Bina, for her time, feedback, references, and information provided.

**Conflicts of Interest:** The authors declare no conflict of interest.

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
