# Peer review of "Integrated Planning: Towards a Mutually Inclusive Approach to Infrastructure Planning and Design"

_land, doi:10.3390/land10121282_

Round 1
Reviewer 1 Report
This is a solid research paper. The topic of relating urban design to infrastructure and engineering is of value to both fields. As an urban designer and professor of urban design, I appreciate the usefulness of the study for professionals and academicians, as well as students.
The example of two dramatically different bridges - one of quite daring design - is a clear and informative case study. The drawings and photos are helpful. More would be welcome, especially more eye-level perspectives and photos. And bit more on the social and physical context. A small map of the city and its location in Argentina would be a good addition.
It might be good for the authors to define the difference between urban design (3D) and urban planning (2D). As they no doubt know, urban design is not an organized profession, and has no guidelines, rules or goals.
"Spatial planning" is not a particularly good or common term in my professional and academic experience. And is ISD a good term?
Participatory design has been around for half a century. As someone who helped found the movement, a little history of it might be good.
Author Response
Dear Reviewer,
Thank you for your time, suggestions and comments. We have addressed these on the manuscript and uploaded a file with a point-by-point description of the changes. Please see the attachment.
Sincerely,
Authors.
|
Reviewer 1 comments |
|
|
Add photos, maps and more on context |
Four new images were placed on the manuscript for this purpose. This was added on the project background and results, location maps and aerial image ( images 1 and 3). Also, a site plan including complementary works was placed on the results (image 19). A pedestiran level photograph, in which the context can be seen, was placed on the results as well (image 20).
|
|
Differentiate between planning and urban design |
This was done in the introduction. Indeed the terms “spatial planning” or ISP are actually a bit contradicting. We changed the title and the concept thorughout the manuscript to make the diferentiation more clear. |
|
Reconsider the notion “spatial planning and ISP”
|
This was done in the introduction |
|
Include a bit more on participatory design. |
This was done in the introduction |

Reviewer 2 Report
Paper title: “Integrated Spatial Planning: Towards a mutually inclusive approach to Infrastructure Planning and Design”. The authors focus on the role of integrated spatial planning, construction engineering, and design in the planning and design process. I find this article interesting and important to the field of urban planning and design, however, I have several suggestions for improvement of the manuscript:
1.
The literature review is relatively narrow and misses some key aspects. For example, there is a significant body of literature relating to integrated spatial planning, construction engineering, and design and evaluation of diverse aspects affecting the quality of the urban design, that the author did not include and should have been referred. For example:
Shach-Pinsly, D., & Capeluto, I. G. (2020). From Form-Based to Performance-Based Codes. Sustainability, 12(14), 5657.
Carmona, M., & Sieh, L. (2004). Measuring quality in planning: managing the performance process. Routledge.
Baker, D. C., Sipe, N. G., & Gleeson, B. J. (2006). Performance-based planning: perspectives from the United States, Australia, and New Zealand. Journal of Planning Education and Research, 25(4), 396-409.
2.
The paper offers a reasonable literature review regarding planning plans in South-Africa, moreover, the authors examine problems that only concern South-Africa. There is no insight into the global nature of the problems that should be addressed in the article more widely.
3.
Line 87-88: “indicators of successful public spaces (accessibility, uses, comfort and image)”
There are no justifications or references indicating how (or why) the authors identified the defined indicators of successful public spaces. There are additional defined indicators for successful public spaces such as: walkability, visibility, sense of security etc. I encourage the author to provide the justifications or references for the defined indicators.
4.
P.3 Lines 96-112:
There is a need to reorganize the “Project background”. The current description is confusing. Passed planning, present planning and future planning are confusing. There is a need to reorganize this section.
5.
Page 4:
There is a description of the planning process, however, the main theme of planning evaluation and integrated spatial planning and design in the planning and design process is not well understood in this section. Please provide a coherent explanation to this main aspect which is crucial to the article.
Author Response
Dear Reviewer,
Thank you for your time, suggestions and comments. We have addressed these on the manuscript and uploaded a file with a point-by-point description of the changes. Please see the attachment.
Sincerely,
Authors.
|
Reviewer 2 comments |
|
|
Expand the literature review |
This was done in the introduction and the discussion. We added anumber of references and all the references suggested. |
|
There is no insight into the global nature of the problems that should be addressed in the article more widely. |
We included this in the introduction, discussion and the final conclusions. |
|
There are no justifications or references indicating how (or why) the authors identified the defined indicators of successful public spaces. There are additional defined indicators for successful public spaces such as: walkability, visibility, sense of security etc. I encourage the author to provide the justifications or references for the defined indicators. |
We have added this under the “methods” section. We added a chart (page 6, figure 6) which contains the criteria for evaluation of alternatives, expanding on additional aspects of public spaces that were taken into account in this case. |
|
There is a need to reorganize the “Project background”. The current description is confusing. Passed planning, present planning and future planning are confusing. There is a need to reorganize this section.
|
We re-organised the description of the project background. We removed figure 2, showing previous proposals of 1999 and 2010 as they created further confusion (we still mention these proposals). We also placed the project background before the methods section. We believe it helps for the reading flow. The process was indeed conplex, as there were proposals of previous years, and within this last process a number of alternatives (high level proposals) were also considered. |
|
There is a description of the planning process, however, the main theme of planning evaluation and integrated spatial planning and design in the planning and design process is not well understood in this section. Please provide a coherent explanation to this main aspect which is crucial to the article
|
We added reference to and a discussion of the planning and urban design process, as well as planning and design evaluation in the introduction and discussion sections. Aditionally, Figure 6 on methods, explains the criteria used for evaluating alternative proposals. |

Round 2
Reviewer 2 Report
The authors have addressed the issues raised by the reviewers and I am satisfied that the paper can proceed to the publication status.